# Towards Safe Robotic Agricultural Applications: Safe Navigation System Design for a Robotic Grass-Mowing Application through the Risk Management Method

José Carlos Mayoral Baños [1,*], Pål Johan From [1] and Grzegorz Cielniak [2]

1   Faculty of Science and Technology, Norwegian University of Life Sciences (NMBU), 1430 Ås, Norway
2   Lincoln Institute for Agrifood Technology, University of Lincoln (UoL), Lincoln LN6 7TS, UK
*   Correspondence: jose.carlos.mayoral.banos@nmbu.no

**Abstract:** Safe navigation is a key objective for autonomous applications, particularly those involving mobile tasks, to avoid dangerous situations and prevent harm to humans. However, the integration of a risk management process is not yet mandatory in robotics development. Ensuring safety using mobile robots is critical for many real-world applications, especially those in which contact with the robot could result in fatal consequences, such as agricultural environments where a mobile device with an industrial cutter is used for grass-mowing. In this paper, we propose an explicit integration of a risk management process into the design of the software for an autonomous grass mower, with the aim of enhancing safety. Our approach is tested and validated in simulated scenarios that assess the effectiveness of different custom safety functionalities in terms of collision prevention, execution time, and the number of required human interventions.

**Keywords:** agrirobots; risk assessment; safe robotics

## 1. Introduction

The enhancement of a vehicle to navigate autonomously in an environment has been studied, for which more applications focus on intelligent behaviours through the addition of robust navigation systems [1]. These approaches require human-robot interaction features that lead to a safety culture, and their design must take into account several aspects to reach optimal performance without any casualties.

As a result of constant and intense research, autonomous vehicles are now becoming part of our present, granting a greater quality of life. Whenever there exists a direct interaction between autonomous devices and humans, there are safety concerns that must be handled.

In the case of agricultural scenarios, agriculture encompasses constraints like any other open-field or human-robot collaboration scenarios, that must be part of the design of the human-aware navigation. Human-robot collaboration in real-world applications has increased since the early 2000's [2], and the goal of a robot-human interaction system [3] includes optimizing the technology benefits.

Safety implies a reliable performance and the reduction of injuries related to the use of agricultural vehicles, mainly tractors [4]. In that context, the ability to detect humans on a field represents one of the core features of robust safety systems [5]. This demand becomes more evident when a robot carries powerful tools.

The risks-related methods are systematic processes for identifying hazards and evaluating any risks, then implementing control measures to remove or reduce them which provide a proper pipeline to address the robotic safety navigation process. These methodologies are a perfect match for autonomous applications where a set of circumstances can lead to potential failures or accidents.

Therefore, the possibility of designing a system with a risk assessment becomes even more tangible to avoid endangering persons and animals, destroying private properties or inducing collateral damage.

In the current work, we present a risk analysis method of a grass-mowing process that was used to design a customized risk mitigation plan and each of the safety functions developed for the safe navigation process for agricultural purposes. The specific contributions of this paper are:

- Design of a safe navigation architecture through the risk assessment methodology for an automated grass-cutting application, including a hazard detection and risk mitigation plan;
- Development of safety functions for open-field agricultural tasks, specifically grass-mowing;
- A safe navigation system designed according to the risk assessment method focusing on human detection and risk mitigation, and designed specifically to handle potential human-robot accidents.

## 2. Related Work

Navigating on unstructured dynamic environments, such as open fields, requires a more robust perception sub-system able to understand the environment dynamics [6]. Therefore, obtaining a comprehensive safety solution for autonomous navigation is a highly complex task, given the broad range of fields, systems, and tasks involved. In addition, there are safety requirements for every application that may vary depending on the application and environment, as industrial and domestic settings may entail different types of hazards that must be addressed to ensure optimal performance.

Standard ISO10218 [7] identifies four classes of safety in robotics systems, and each class uses different methods for safety assurance: control, motion planning, prediction, and psychological considerations [8,9].

There are human-aware navigation solutions available that are based on several people detection algorithms and considerations, for which it is possible to consider a safety state as a fault-tolerant system [10] (control), or use human trajectory prediction for collision avoidance [11] (prediction), or even improve with the addition of safety rules applied to occluded obstacles [12] (motion planning).

In the field of autonomous navigation, safety is a critical consideration to prevent accidents and ensure the proper functioning of the systems. One approach to addressing safety concerns is to develop a taxonomy of safety features. For example, Zacharaki et al. [13] proposed a safety feature taxonomy that includes various aspects of autonomous systems. Specifically, the taxonomy identifies perception, cognition, action, hardware, social and psychological features, and a hazard analysis as critical components for reducing the risk of accidents or failures.

As safety is a complex term to define, an attempt to standardize the implied requirements in agricultural applications comes within the International Standard ISO18497 [14] that summarizes safety as a function of human and obstacle detection, safe-state transition procedures, the suppression of unintended excursions, a remote emergency stop mechanism, and the non-presence of humans and obstacles in a defined hazard zone. Additionally, other standards, such as ISO 31000 [15] and ISO 21000 [16], demonstrate how to complete a risk assessment and manage successfully.

Standard 18497 [14] explains that human safety is one of the fundamental aspects required in agricultural machinery where people detection can be considered as the main feature of an autonomous device in an agricultural task.

People detection has involved the development of a large set of solutions over the years. A very popular solution is to use cameras to detect people via feature extraction algorithms, from the well-known extractors SURF (speeded-up robust features) or SIFT (scale invariant feature transform) to hog (histogram of oriented gradients) solutions [17]. However, in recent years, machine learning and deep learning-based approaches have

achieved overwhelming results for object and people detection. For example, Redmon proposes a new deep learning architecture called YOLO, that stands for "you only look once", and its third version was released on 2018 in [18]. The main characteristics of this architecture rely on its input as an entire image, and its output gives a set of bounding boxes in the same frame. It is also robust for object overlapping.

Another sensor used for human detection in real-time applications is the 3D LiDAR, where a combination of filtering and convolutional neural network layers can detect humans in cluttered environments [19], or with a two-stage deep learning architecture, such as PointRCNN, developed by Shi et.al. [20], that decomposes the problem into bounding box proposal generation with a refining coordinate process as the last stage.

For safety systems in agricultural environments, a camera is used in different manners, highlighting collision avoidance, human intention recognition, and distance to robot metrics [21]. Similarly, Raja et.al. [22] developed a customized costmap used for smart navigation avoidance during an agricultural task that avoids stepping on healthy crops.

Indeed, robotics has evolved from a stage where safety was handled by reactive mechanisms to decision-making processes where safety policies play a major role in the decision-making. With the use of more advanced sensors, algorithms, and machine learning techniques, modern robots are capable of making informed decisions to ensure safe and efficient operations [23].

Designing safety policies in the system design stage reduces the hazardousness of robotic applications. Woodman et.al. suggested a high-level system analysing the hazard sources to achieve real-time safety monitors [24]. Additionally, a risk-based method is used to integrate safety requirements implementing a safety-driven controller through the definition of safety policies [25]. The decision-making process is also implicit in a safe autonomous navigation task. For instance, once a mobile robot obtains a person's position, it might decide what to do with this information. For example, Must and Lauderbaugh [26] have provided an offline safety-based architecture, calculating the reliability of a plan to avoid safety constraints being violated. Using a statistical evaluation, the solutions analyse the feasibility of a generated plan before being executed.

Another implementation for real-time applications is with a safety framework called SMOF [27] that has been developed to generate rules that model a robot as a fault-tolerant system, and focuses on monitoring safety rules through the inclusion of active monitors. The framework considers three basic system states: safe, warning, and catastrophic.

Schratter and colleagues have reformulated the problem as a partially observable Markov decision process (POMDP) [28], that extends the capabilities of a safety system merging an autonomous breaking system into the decision process. This approach assumes a pedestrian's constant velocity and predicts the next positions of the human and the time to collision, improving the system's safety capabilities.

It can be derived that this is due to the artificial intelligence's high performance, that is well-known in robotic-related tasks. Safe navigation and risk management processes can be designed as a combination of fuzzy logic, risk management, and deep learning [29].

Risk estimation is the final stage of the risk management process. The use of different metrics in human-robot collaboration stages should be a system priority that relies on robot perception. These metrics can be based on social cues, visiting points to define the riskier spots in a map, or in the form of a multi-variable cost function [30]. Further, an analysis of a robotic navigation system concludes that in applications where safety is a concern, additional considerations must be monitored as the CPU load or the performance, and detected anomalies should be input into the safety system [31].

In this paper, we suggest the explicit integration of the risk management into the autonomous navigation process, and evaluate its performance in a simulated environment of a grass cutting task where human-robot collisions are unwanted. Hereby, all of the assumptions and considerations are explained, describing a custom autonomous solution for the mentioned task to make it as safe as possible.

## 3. Problem Statement

For an autonomous application, it is mandatory to understand the limitations and characteristics of the task to be automated. The test case for our research is an autonomous grass mower under the scope of the GrassRobotics project.

The main core idea is to increase the productivity of the grass-mowing and collecting process, and decreasing soil compaction as a natural consequence of using lighter vehicles for such applications. The selected robot Thorvald [32] (Figure 1) consists of a four-wheeled robotic platform, and each wheel is controlled by two electric motors for steering and accelerating. The tool is mounted on the front part of the robot to increase control over path planning.

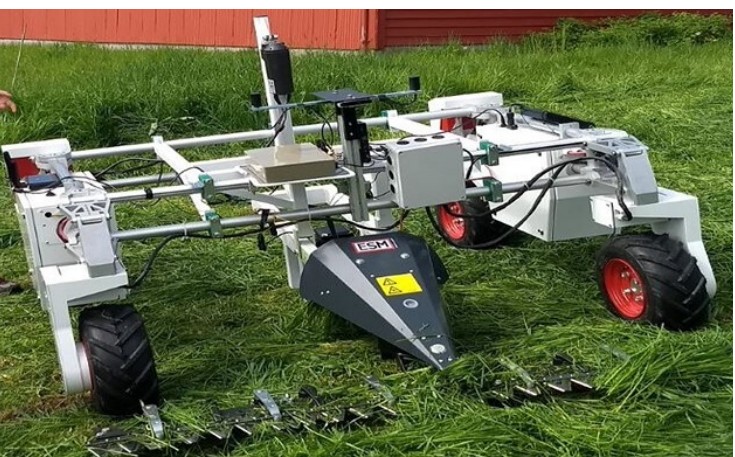

**Figure 1.** Our robotic platform.

Our sensor set consists of: an RGBD camera, a VLP-16 Velodyne LiDAR, an IMU, and a set of RTK antennas. The robot also carries two computers and a WiFi router. The platform used is designed to carry tools for agricultural applications.

In Table 1, some task properties are mentioned at three description levels: environment, task, and platform. This information is of major importance to keep track of the resources and to define the requirements of the system in such a way that the safe navigation architecture constrains its execution within the available resources. The task description table summarizes the environment characteristics in which the robot normally runs, including the weather conditions and seasonality.

The components selection and features are based on the problem statement shown in Table 1, using this information to select the best configuration for our system. The selected system's design makes allowances for the grass-mowing application with three modes: crop-following, line changing, and free navigation. The system configuration is shown in Table 2, introducing the main components for each navigation mode.

**Table 1.** Task description.

|  | **Property** | **Value** |
|---|---|---|
| Environment | Structured | No |
|  | Season | Spring-Summer |
|  | Weather | Sunny |
|  | Collaboration | Required |
|  | Multi-Agent | No |
| Mowing | Material | Grass |
|  | Nominal Speed | 3.35 mph |
|  | Tool | Industrial Cutter |
| Platform | Power | Electricity |
|  | Drive | Omnidirectional |
|  | Name | Thorvald |
|  | Sensor Set | Customized |
|  | Wheels | 4 |
|  | Motors | 8 |

**Table 2.** Navigation architecture design.

|  | **Selection** | **Mode** | **Layer** | **Avoidance** |
|---|---|---|---|---|
| Nav. Stack | MBF | ALL | None | Enabled |
| Motion Planners | Global Planner | Line Changing | Free | Custom |
|  |  | Free | Obstacle | Enabled |
|  | Carrot Planner | Crop Following | Custom | Disabled |
| Local Planners | DWA | Free | Custom |  |
|  | MPC | Crop Following | Custom | Custom |
|  |  | Line Changing | Obstacle | Enabled |
| Localization | RTK-GPS and Odometry | ALL | - | - |
| Mapping | Custom | ALL | Custom | - |

## 4. Approach

For this research, a safe navigation application requires merging two main aspects that frequently appear in the literature: risk management and autonomous navigation. Both systems require an improvement at the design stage that analyses all of the safety requirements, and they must be analysed before proceeding to development. This step is of major importance to handle further potential risks throughout the system's lifetime.

### 4.1. Risk Management

As a start, we consider the safety system as a five-step risk management process based on [16,33] and modified for handling run-time applications using different risk policies, as shown in Figure 2. This figure summarizes the approach followed in the present research, highlighting the importance and advantages of mapping the potential risks into the design step.

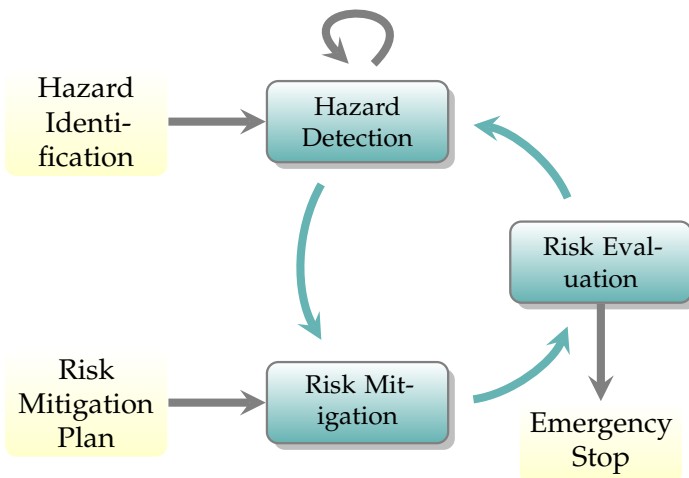

**Figure 2.** Risk management plan.

It starts with a hazard identification step that provides all potential hazards. This information is sent to a detection phase that is constantly waiting for these events to happen in the scene. Later on, a risk mitigation plan links this to an action according to a predefined plan. Finally, the system evaluates if the previous step output is as expected and whether the risk is acceptable or not. If the risk is high, the robot must stop all actions to avoid any casualties.

The hazard definition starts by defining the scenario and task that the robot will perform. In the case of this research, the tasks run in open-field environments and the device executes agricultural applications, such as seeding, harvesting, cutting, or sowing. Given these elements, the full list of hazards found for this task is provided in Table A1 in Appendix A. However, due to the complexity of the nature of the experiments presented in this paper and the early status of the robotic platform, a subset of this is picked to be included in the design stage (Table 3).

**Table 3.** Hazards definitions.

| Risk ID | Hazard Definition | Severity | P.Oc. |
|:---:|:---:|:---:|:---:|
| 01 | Living Being Not Detected | S0, S2 | E4 |
| 02 | Living Being Detected in Proximity | S0–S3 | E4 |
| 06 | Trajectory Intersects Human Trajectory | S1–S3 | E4 |
| 09 | Running out of Borders | S2 | E3 |

Table 3 covers the most common hazards in agricultural applications mentioned in ISO Standard 18497 [14]. However, all of these are caused by different errors or failures. Therefore, a failure mode effect analysis (Table A2 in Appendix B) provides a deeper understanding of the causes.

Our FMEA splits into low-level and high-level failures and effects. The high-level failures represent the process modules and the low-level represent the ones caused directly by sensing failures. Once the hazards are defined, the cyclic detection stage can perform as expected. All true positive-detected events are managed by the risk mitigation plan.

The risk mitigation plan consists of two steps. First, it defines some risk policies that could potentially address all risks together, and then it links all policies to a specific risk. This process ends in a customized risk plan for agricultural applications based on the hazards defined in Figure 2.

The hazards definition aims to guide the risk policy development. However, either detecting or not detecting living and non-living beings in the scenes can result in a different severity range according to the direction of the robot.

Therefore, a risk zone definition has been created to handle these situations defined per the policy. The summary of the proposed mitigation plan is described in Table 4.

**Table 4.** Risk mitigation through proposed policies.

| Risk ID | Hazard Definition | Severity | Function | Type |
|---|---|---|---|---|
| 01-A | Living Being not Detected in Lethal Zone | S0 | State-Transition | Event |
| 01-B | Living Being not Detected in Danger Zone | S1 | State-Transition | Event |
| 01-C | Living Being not Detected in Warning Zone | S2 | State-Transition | Event |
| 01-D | Living Being not Detected in Safe Zone | S3 | State-Transition | Event |
| 02 | Living Being Detected in Proximity | * | Advance-Proximity | Continuous |
| | | | State-Transition | Event |
| 03-A | Non-Living Being not Detected in Lethal Zone | S0 | State-Transition | Event |
| 03-B | Non-Living Being not Detected in Danger Zone | S1 | State-Transition | Event |
| 03-C | Non-Living Being not Detected in Warning Zone | S2 | NONE | Idle |
| 03-D | Non-Living Being not Detected in Safe Zone | S3 | NONE | Idle |
| 04 | Non-Living Being Detected in Proximity | S3 | Basic-Proximity | Event |
| | | | Collision-Avoidance | Continous |
| 05 | People Laying on the Grass | * | State-Transition | Event |
| 06 | Trajectory Intersects Human Trajectory | * | Advance-Proximity | Continuous |
| 07 | Injured Animals on the Crops | * | Basic-Proximity | Event |
| | | | Collision-Avoidance | Continuous |
| 08 | Tool Malfunction | S2 | Monitor | Continous |
| 09 | Running out of Borders | S2 | Knowledge | HI |
| 10 | Encoder Malfunction | S1 | FMEA | HI |
| 11 | Camera Malfunction | S2 | FMEA | HI |
| 12 | LiDAR Malfunction | S2 | FMEA | HI |
| 13 | GPS Malfunction | S2 | Knowledge | HI |
| 14 | Speeding | S2 | Advance-Proximity | Continous |
| | | | Knowledge | FMEA |
| 15 | Communication Lost | S2 | FMEA | Idle |
| 16 | Others (to be defined) | | | |

* means applicable from S0 to S3.

Section 5 provides the details of the different risk policies mentioned in Table 4. This strategy guides the actions run by the robot to reduce all danger for the people and obstacles within the environment it performs the task. Once each action is executed, the system evaluates the risk, and if it is above a defined threshold, the robot must stop its task.

*4.2. Autonomous Navigation*

Additionally, safe navigation is an autonomous execution pipeline that implicitly embeds safety into its major design considerations. However, explicitly integrating safety into autonomous navigation requires special considerations.

For the purpose of this research, safe navigation implies that safety is the core function instead of the execution module, as normally considered. Our conceptual definition of safety navigation is based on the considerations wrapped in the International Standard ISO18497 [14]. To fully consider a system safe, the system must include four different modules. Our definition considers explicitly four modules to guarantee a robust focus on safety: monitoring, communications, execution, and safety systems. Together, the four provide a robust system deployed in different software levels ranging from low to high. A brief definition for each subsystem is:

- **Execution:** This module handles the task execution, including localization, mapping, trajectory planning, and control. The agricultural robots normally localize themselves by GPS-based solutions. In mapping approaches, there are two main tendencies using a static map or map-less solutions. A high number of controllers and planners can be used to make the robot execute a given task; however, the selection is influenced by the agricultural application's characteristics. The execution module must stop whenever the safety module is not working;

- **Communication:** The second module addresses the problem of creating human-robot communication. This can include visual, speech, haptic, or smartphone app interfaces to receive and send information to the robot. In this module, the concept of human intervention is introduced, enabling a safe human-robot interface;

- **Safety:** This monitors the external events that interrupt the robot's safe state. Any object or human that is put at risk of collision, injury, or death must be detected in order to prevent these from happening. This, and the execution module are highly dependent;

- **Monitoring:** Any of the other systems can fail during run-time. The last module watches the correct behaviour of the other three modules. This module can interrupt the robot's task.

Execution and communication tasks are highly developed, as demonstrated in the literature. However, the safety module is normally implicitly considered in perception systems. These elements take into account simple policies based on precision or on a number of retries. However, more robust safety systems can be deployed if the task is fully understood and a hazard analysis is carried out before the development phase. For the purpose of the current work, the definitions of execution and safety systems are hereby established.

4.2.1. Execution Module

There are some navigation frameworks available on the market, such as move_base, move_base_flex [34], and Nav2 [35], that have been developed in the popular robotic framework ROS. All of them rely on a four-step navigation procedure that comprises: localization, mapping, path planning, and path execution. However, Nav2 provides, by default, a map-independent navigation framework that makes the navigation stack modular and flexible.

The grass-mowing process is an iterative operation; however, the flexible design of the Nav2 and the behaviour tree structure might be too complex for the problem per se. In addition, the Thorvald platform software is developed on ROS1 framework. The three modes described in Table 2 represent the main action in the grass cutting procedure.

Free navigation runs the default collision avoidance navigation system used to go in a defined start position, grass collector, and other coordinates. Crop-following is a set of global and local planners with parameter values set for such behaviours. A customized agricultural task costmap based on [22] constrains the planners to avoid undesired behaviours. Finally, row changing defines an MPC for the differential drive to turn in tighter curves than an Ackerman drive. The customized costmap is explained in Section 5.4.

### 4.2.2. Safety Systems Module

Our safe navigation system uses the risk management process for detecting and handling any detected hazard during the execution. This identification stage is of great importance for our system because it provides a guideline for our autonomous system.

## 5. Safety Functionalities

Safety functions must solve a specific safety objective. In the case of the grass-mowing task, human safety is a major priority for the optimal autonomous execution of the task. Therefore, our main focus remains on this critical aspect analysing different safety functions, and how they impact safety without compromising performance.

As an additional classification for our functions, we defined their classes as continuous and event-based. The first is executed continuously while the robot is active, meanwhile the event-based is called once per detected hazard. Finally, HI stands for human intervention and it calls for the emergency stop and requests assistance.

### 5.1. Basic Proximity Function

This basic policy is nothing other than a simplified approach for the safe action selection in robotics using the Euclidean distance between the robot and closest object or person. This approach assumes that the robot needs to perform a safe action for returning to a safe state and uses prior knowledge to select an appropriate action based on the distance to the nearest object or person. A simple visualization of our approach is given in Figure 3.

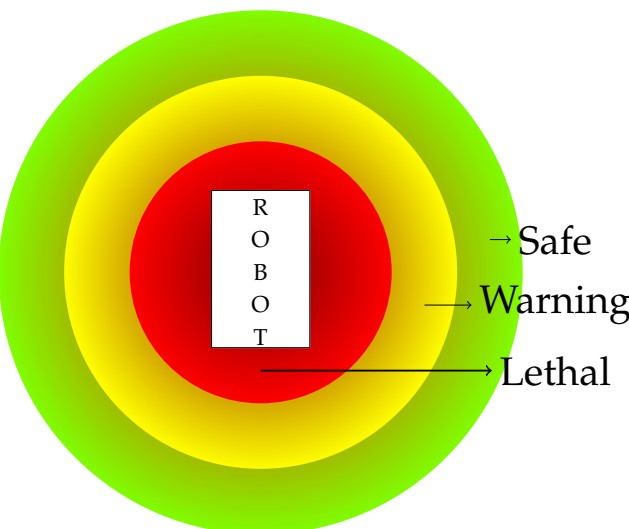

**Figure 3.** Basic proximity safety function.

The safe action execution also includes a set of safety rules and their only purpose is to minimize true positives and negatives from the object detection algorithms based on logic:

1. Once the hazard state is reached, the robot cannot transit to the safe state directly, i.e., a person is no longer detected in the hazard zone;
2. If the detection algorithm does not provide feedback after a timeout, it might mean that the robot does not understand the environment (blind robot), therefore, the robot must stop and/or require human assistance;

3. Safety system acts as a navigation monitor, therefore, it interrupts and commands the system if any of the safety rules are threatened.

*5.2. Braking System Function*

Real-world is a complex and dynamic environment where obstacles in static conditions are commonly found. Then, object tracking represents a good choice for handling a braking system policy with tracking functionalities.

Risk estimation for the environmental obstacles can be modelled as a function of time, position, the robot's orientation, and risk layers around the robot. Mayoral et al. [36] provided a framework for risk assessment using, as a test case, human detection in an open-field environment, and its conceptual idea can be observed in Figure 4.

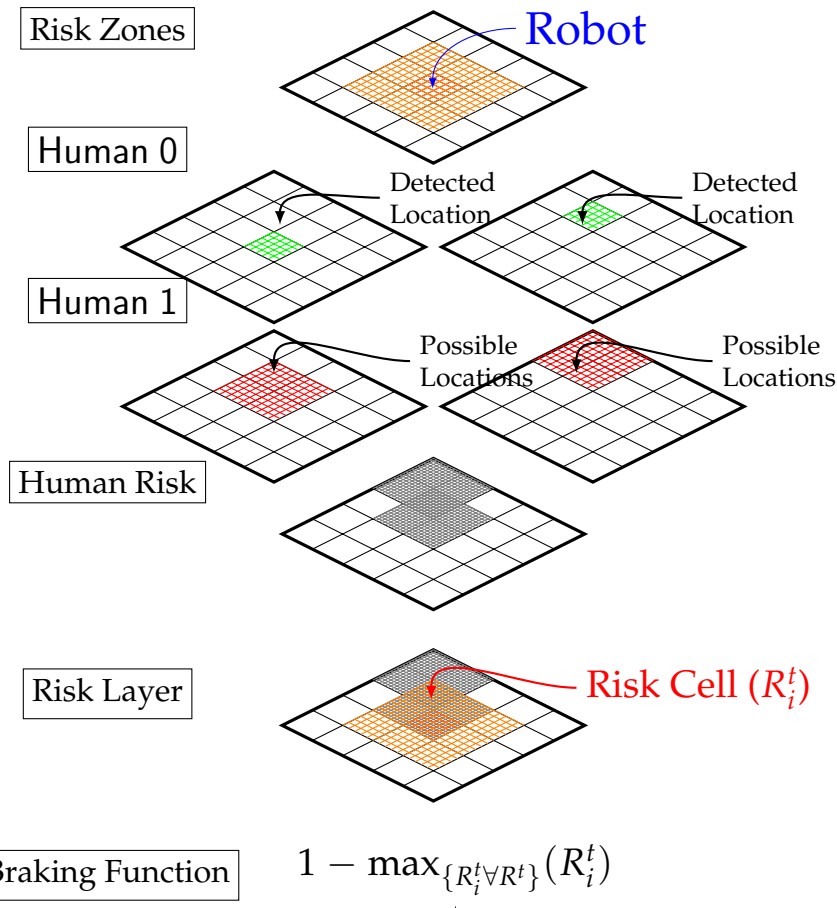

**Figure 4.** Braking system function.

This method is used as a risk-based braking system, and its concept utilizes human 3D position sampling to evaluate the 2D projected position in the map using a degradation function. Finally, it concatenates all of the layers and injects the resulting risk index as a braking parameter into the motion controller. Those features make this policy a good choice for a continuous risk policy where the depth-limited sampling mitigates the processing power and time to generate a run-time solution.

*5.3. Human Risk Assessment Function*

With AI's steep development, detecting obstacles [37,38] and vegetation [39] for open-field agricultural applications has been developed in a variety of sensors, including cameras and LiDARs with a large set of solutions that fulfil extraordinary results. However, the risk evaluation process comes from a set of safety rules that processes the robot's state to identify the next action.

An approach to classify human risk according to a single RGB frame is presented in [40], this reuses a one-shot neural network classifier to estimate the risk according to the distance from a camera without using depth information. The idea is simple as the closer the person moves to the camera, the greater the risk and the lower the class index, according to the original research.

For this risk assessment policy, the previous approach is extended and taken as a state transition. This method attaches a single action according to the transition between the previous and current state. To achieve this purpose, this policy encompasses a definition for the current risk state that uses all human risk classification $P_{class}$ at time $t$:

$$R(t) = \min_{p_i^t \forall P_{class}} (p_i^t)$$

The policy's action criterion designates the risk mitigation operation to minimize the risk. In the start time, the state is set as unknown to mitigate a potential risk in the initial state. State transitions should be sequential, where a transition higher than one level change leads to a mitigation action. A complete risk mitigation action must be defined by the system designer, and Table 5 displays ours.

**Table 5.** States transition safe action selection.

| $t-1$ \ $t$ | Lethal | Danger | Warning | Safe | Unknown |
|---|---|---|---|---|---|
| Lethal | Stop | Resume | Resume | H.I. | Stop |
| Danger | Stop | Stop | Resume | H.I. | Stop |
| Warning | Stop | Stop | Idle | SpeedUp | Slowdown |
| Safe | H.I. | H.I. | SlowDown | Idle | Slowdown |
| Unknown | H.I. | Stop | Resume | Resume | Idle |

This risk assessment policy is designed for this particular approach every time an action brings out the following effects in the system:

- Idle: Sets current state $t$ with previous risk state $t-1$;
- Speed-Up Enables full speed;
- Slowdown Constrains mobile robot by limiting maximum speed to a safe speed $t^*$;
- Stop: Pauses given task, motion, and shuts down power tools;
- Resume: Resumes current task, motion, and shuts down power tools;
- HI: Requests human intervention, and leaves the decision-making process to a human operator.

The significance of human intervention in monitoring the autonomous devices cannot be overstated. The HI action is designed to detect any inconsistencies in performance, upon which the robot must immediately stop and alert the operator. The human operator is then able to decide whether to allow the task to continue, following which the robot can resume its operation.

*5.4. Knowledge-Based Function*

This safety policy is developed through a special costmap representation that will avoid trespassing on the workspace. A representation for a four-row length field is shown in Figure 5, where the blue regions identify the acceptable region that the robot must be located in to mow the grass, and the green indicates the limits of the workspace. It is a simple idea that provides feedback to the robot when it trespasses in an area where it should not be.

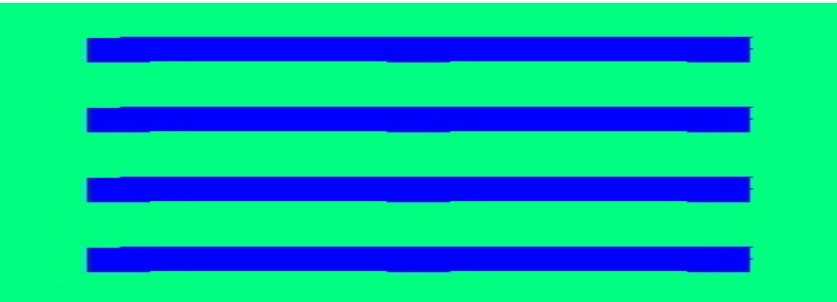

**Figure 5.** Simple task knowledge costmap representation.

This simple smart costmap will monitor, in a fast manner, the motion of the robot in the crop following mode. In addition, it will also crash the local planner when the localization of the robot is out of the blue region (acceptable localization region). Even the global planner is set up to fail when the robot is out of the green region, allowing for a correct determination of the running out of borders hazard.

## 6. Navigation Architecture

The safe navigation architecture proposed for this research is shown in Figure 6, and it consists of four actions and three components. Those components are the map manager, the navigation manager, and the safe controller.

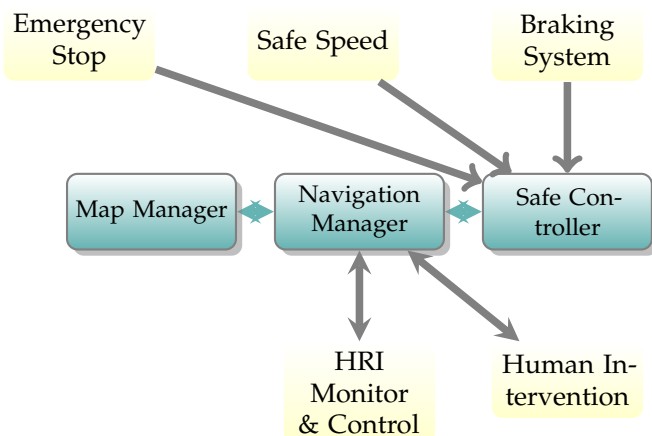

**Figure 6.** Navigation architecture.

The map manager sets or resets the workspace map (Figure 5) according to the intended action of the robot, for example, tool on or off. The navigation manager commands this action and manages the execution of the task and the controller selection. The HRI monitor and control component receive the desired task from a human operator who is also responsible for resetting the human intervention that stops the robot until a confirmation is received. Finally, the safe controller computes the final speed through the aggregation of all of the speed signals.

## 7. Simulations

Agricultural robots perform mostly under open-field scenarios where people work within their limits and sharing space with the robots. Therefore, our experiments mimic this situation by testing the performance of our approach.

The experiments are designed to perform a risk assessment in a simulated-based scenario where all hazards described in Table 3 can be tested without endangering humans or private property. The simulated environment looks like a field where any agricultural application could be performed (Figure 7), and it also consists of two automatized humans moving at different speeds and in different trajectories that were developed on the

Gazebo simulator. Our simulation engine commands the trajectory of each of the humans automating their motions, randomly selecting their speed, direction, and distance to move.

The experimental setup uses an analysis for hard scenarios, so it is challenging for any perception system. Thus, there are humans moving around all the time along the environment. There is no prior knowledge of when a collision might happen or when a person is not detected.

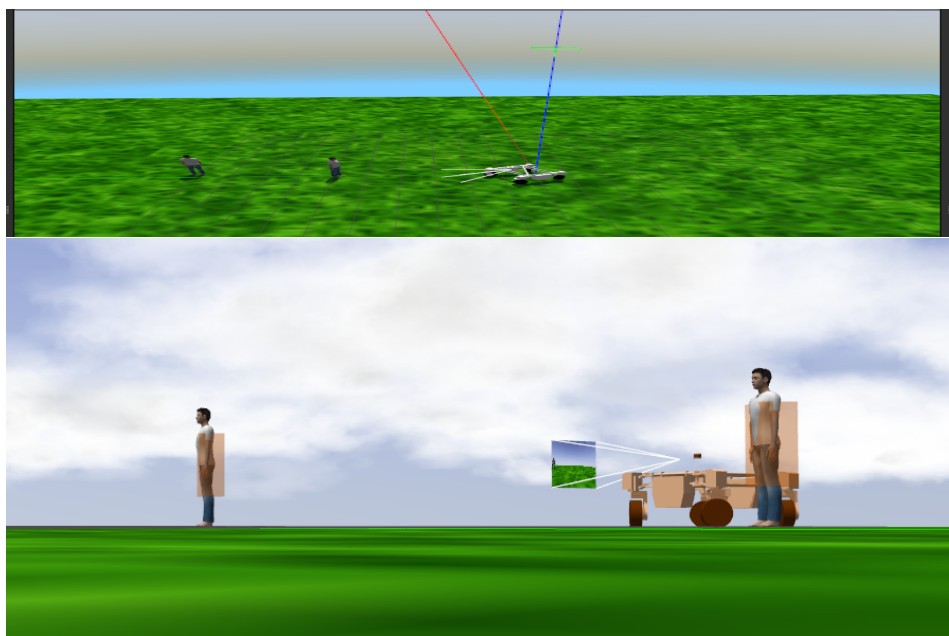

**Figure 7.** Gazebo simulator with an open-field agricultural environment and the collision boxes.

The simulated environment consists of a field-like simulation, measuring $10 \times 10$ m, in which the robot must cover four grass rows. The main objective of the experiment is to evaluate the risk to humans during the execution. The two moving persons possess no intelligence to avoid the robot; therefore collision avoidance relies entirely on the safety functions. We define five experimental setups, the base results with no safety function component, and four with one or two different safe functions. All of our experiments are set in challenging conditions simulating persons within the robot's workspace. The human agents re-spawn every 15 s in simulation time to test the robustness of our system.

The experiments consist of five test cases, including one used as a reference (no safety system). For each test case, we performed twenty runs over a simulated field under the same challenging conditions. The main measure of the experiments is the number of collisions between the robot and human. A collision is considered genuine only if it occurs in front of the robot while it is moving at a speed greater than 0.1 m/s. Both humans are re-spawned every time there is a collision, or when there is no collision affecting its trajectory after three motion repetitions.

Our first case and reference run without any safety systems and it is used as a reference to analyse the performance of the other cases. The other four experiments provide stochastic comparisons with collisions with humans, the average time between collisions, speed, and interventions. Each test case performed is monitored with one safety function before it is explained, and they are:

1. No Safety;
2. Basic Proximity;
3. Braking System (based in [36]);
4. Human Risk Assessment (based on [40]);
5. Functions Fusion (Advance Proximity and State-Transition).

In addition, for all test cases, the knowledge-based function (Section 5.4) ensures that the robot with the tool never leaves the workspace. Together with a motion planner, the robot must pass the same path over and over without any path modifications.

For evaluation, there are two main metrics: the mean execution speed per cycle and the average execution time. Additionally, safety components, such as the number of collisions, mean time between collisions (MTBC), and mean time between interventions (MTBI), complement the assessment. Furthermore, we also calculated a percentage improvement in terms of MTBC (%Imp), that grades the average time between collisions during the execution.

## 8. Results

The experiments analyse the performance of the presented safety functions for avoiding fatal collisions in a human-robot collaboration environment. The experiment's main goal is to prove the robustness of our system and provide a benchmark for future research. Table 6 summarizes the findings of the five test cases.

**Table 6.** Performance analysis for the five test cases.

|  | No -Safety | Basic Proxim-ity | Braking System | H. Risk Assess-ment | Functions Fusion |
|---|---|---|---|---|---|
| $\overline{V_x}$ | 0.32 | 0.11 | 0.24 | 0.28 | 0.26 |
| $\overline{T}$ | 160.5 | 476.8 | 234.9 | 184.9 | 222.1 |
| #Col. | 55 | 30 | 28 | 31 | 19 |
| MTBC | 101.5 | 442.1 | 327.6 | 222.6 | 417.8 |
| MTBI | - | - | - | 45.16 | 52.90 |
| %Imp. | - | 435.5 | 322.7 | 219.2 | 411.5 |

Our baseline provides a performance overview of how a mobile robot would work in a field without any safety system. It shows the highest mean speed along the execution (0.32 m/s), and the longest execution time per iteration as well (160.5 s). In comparison, the basic safety function over performs the baseline in its number of collisions, and its improvement in terms of MTBC is around 435.5%. However, in terms of coverage time per execution iteration, it performs poorly at 476.8 s. Therefore, using this safety function provides a more robust solution, but its performance is severely affected, producing longer executions times.

The human risk assessment function performs in the shortest time and shows an improvement in MTBC of 219.2% and MTBI of 45.16 s. Furthermore, the functions fusion is the one with the fewest causalities, an MTBC of 417.8 s, which is the closest one to the basic function, and an MTBI of 52.90 s.

The simulations run in an environment with two simulated humans randomly moving around; however, the robot's motion is also important to understand whether the robot attempted to decelerate and avoid an accident. The speed profiles of the executions are shown in Figure 8 and display the collision locations on the map. Our safety functions are able to minimize fatal casualties; however, it is not a perfect solution.

The figures shown in Figure 8 display the basic plot of a farm land, where the axes represent the height and width of the terrain on which the robot runs and performs experiments. On the one hand, Figure 8a demonstrates the profile without any attempt at stopping or reducing speed when a risk situation appears in its workspace. On the other hand, when safety functions complement the performance, there is a decrease in collisions, especially when our human risk assessment and braking system are fused.

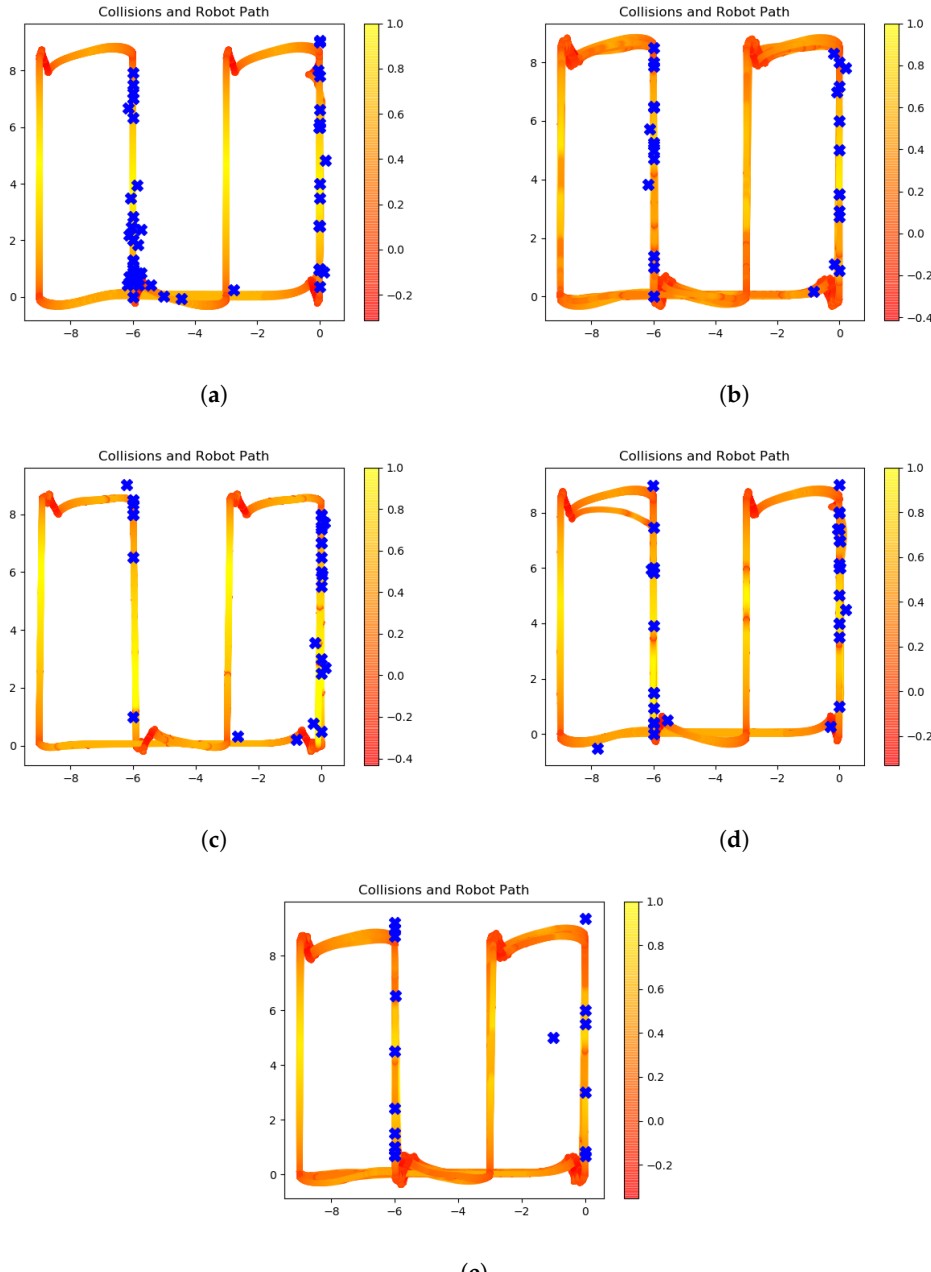

**Figure 8.** Speed profile for safety actions. (**a**) No safety functions. (**b**) Braking system. (**c**) Simple proximity. (**d**) Human risk assessment. (**e**) Fusion test case.

The braking system (Figure 8b) shows attempts for decreasing the velocity when a person is detected. However, the simple proximity function (Figure 8c) struggles to manage the complexity of the environment, showing non-significant slow-down attempts. Our camera-based solution (Figure 8d) seems to perform well to reduce the speed before the collisions, but the reduced vision range avoids having a full range detection; but, once fused, the speed footprint changes along the trajectory (Figure 8e).

Further, in the case of the risk assessment function, the robot calls for different safety actions during the experiments. Ignoring idle cases where the state remains constant for two consecutive times, the actions called distributions are shown in Figure 9, observing that non-consecutive transitions are not common along the execution of a task.

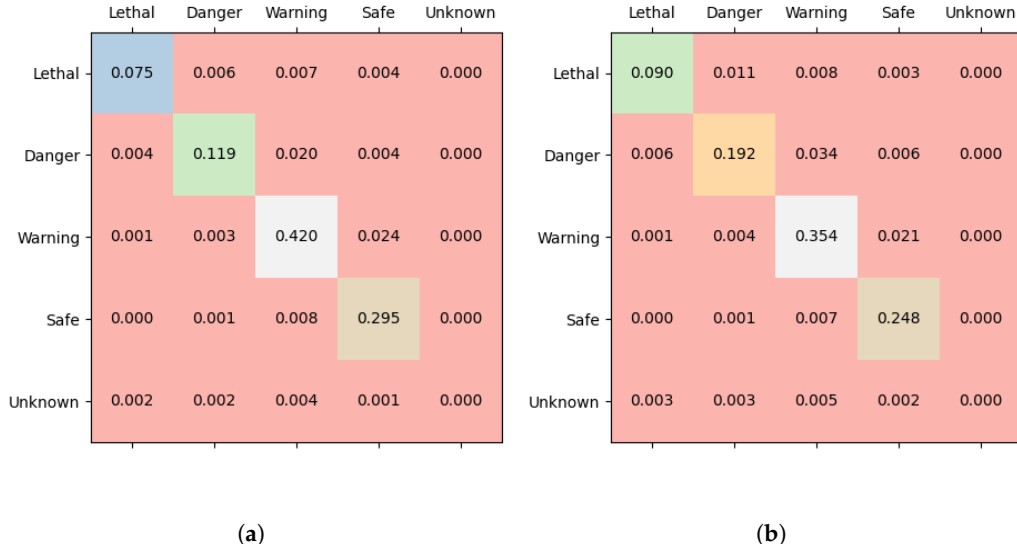

(**a**)                                      (**b**)

**Figure 9.** Action request calling results. (**a**) Transition test case. (**b**) Fusion test case.

The state of the risk has preserved most of the execution time (diagonal values) when the non-diagonal values are the events where the simulated humans change their location in terms of the risk zones. Furthermore, the diagonal values of the figures illustrate the transitions over time of the system for this safety function, demonstrating the tracking behaviour of the people in the robot's scene.

In addition, Figure 9 demonstrates that the safety actions that are performed most frequently are stop and idle (action mapping is shown in Table 5). Meanwhile, transitions from and to the unknown state are not common during the execution.

## 9. Conclusions

The current paper presents an innovative approach to integrating object detection and safety systems in autonomous grass-mowing applications. The primary contribution lies in utilizing a risk assessment method to develop a customized navigation stack that prioritizes safety. Our autonomous architecture demonstrates great performance in handling risks during the execution of autonomous tasks.

To explicitly integrate safety functions in an autonomous mobile application, in this work, we generate a risk mitigation plan that covers a wide range of potential risks for industrial grass-mowing solutions. Risk metrics, including average time per cycle, inform the performance impact when integrating a safety system. Our findings demonstrate a significant improvement in performance, with the fusion case showing an increase from 160.48 to 222.13 s, and a decrease in the number of collisions from 55 to 19. Furthermore, we compared four safety functions in complex situations and found a remarkable improvement of more than 400 percent in MTBC.

Despite these promising results, experiments in production environments are necessary to obtain more accurate metrics and achieve international safety standards in autonomous agricultural applications. Human factors, such as fatigue, technology reliability, and intelligence, must be part of future experiments providing more reasonable and in depth human behaviours.

The current research provides the first steps toward a safe autonomous agricultural robotic grass-mowing application. However, there is a set of additional features that provide a more robust solution for handling real-world applications. Advanced fault-detection algorithms can be employed to complement sensors' accuracy and reliability. In addition, analysing recent networking protocols can expand the range of human-robot interfaces in open-field environments. Finally, incorporating international safety standards into the overall design can address additional hazards and indoor environments.

**Author Contributions:** Conceptualization, methodolody, software, investigation & writing: J.C.M.B.; Conceptualization, supervision, Writing—review & editing: G.C.; Supervision & Project administration, P.J.F. All authors have read and agreed to the published version of the manuscript.

**Funding:** This research received no external funding.

**Conflicts of Interest:** The authors declare no conflict of interest.

## Appendix A

In this section, the complete analysis and description of the hazards are presented. In the case of hazard description, a more accurate interpretation of the potential risk is described in Table A1.

**Table A1.** Hazards definition.

| Risk ID | Hazard Definition | Severity | P.Oc. |
|---|---|---|---|
| 01 | Living Being not Detected | S0, S2 | E4 |
| 02 | Living Being Detected in Proximity | S0–S3 | E4 |
| 03 | Non-Living Being not Detected | S1, S2 | E4 |
| 04 | Non-Living Being Detected in Proximity | S1, S2 | E4 |
| 05 | People Laying on the Grass | S0, S3 | E2 |
| 06 | Trajectory Intersects Human Trajectory | S1–S3 | E4 |
| 07 | Injured Animals on the Crops | S1–S4 | E3 |
| 08 | Tool Malfunction | S2 | E2 |
| 09 | Running out of Borders | S2 | E3 |
| 10 | Encoders Malfunction | S1 | E2 |
| 11 | Camera Malfunction | S2 | E2 |
| 12 | LiDAR Malfunction | S2 | E1 |
| 13 | GPS Malfunction | S2 | E3 |
| 14 | Speeding | S2 | E3 |
| 15 | Communication Lost | S2 | E2 |
| 16 | Others (to be defined) | | |

The previous table provides a guideline for future work of the presented research, to make a more robust approach ready for real-world operations. The hazard list is generic for a grass-mowing agricultural task, and the last line introduces the facility to extend the hazards list or to create a more specific description for another task rather than grass-mowing.

## Appendix B

A failure mode effect analysis demonstrates the relation between potential failures and the recommended actions to prevent catastrophes. The FMEA used for this research is shown in Tables A2 and A3 where high- and low-level failures are divided. Note that some low-level failures, such as sensor-related ones are not part of the current research; however, they must be a core part of future extensions of the system hereby introduced.

**Table A2.** The high-level failure mode and effects analysis.

| Process Step | Potential Failure Mode | Potential Failure Effects | Potential Causes | Current Controls | Recomm. Action |
|---|---|---|---|---|---|
| **High-Level Failures** | | | | | |
| Execution | Localization Imprecision | Speeding | GPS Failure | Monitoring | Slow Down |
| | | Speeding | Odometry Failure | Kalman Filter | Slow Down |
| | | Unintended Excursion | GPS Failure | Monitoring | EStop |
| | | Unintended Excursion | Odometry Failure | Kalman Filter | EStop |
| | Unintended Excursion | Collision | Odometry Failure | Perception System | EStop |
| | Speeding | Collision | Localization Imprecision | Velocity Constraints | Abort |
| | Wrong Path | Unintended Excursion | Localization Imprecision | Kalman Filter | EStop |
| | | Collision | Safety System Error | Perception System | Abort |
| Safety | Collision | Injuries/Death | Wrong Path | Perception System | ESTOP + H.I. + Shutdown |
| | | Destroy Property | Wrong Path | Perception System | ESTOP + H.I. |
| | Safety System Error | Injuries/Death | Camera Failure | Perception System | ESTOP + H.I.+ Shutdown |
| | | Injuries/Death | LiDAR Failure | Perception System | ESTOP + H.I. + Restart |
| Comms. | App Comm. Lost | Injuries/Death | Bluetooth Failure WiFi Failure | Monitoring | Abort + H.I |
| | Visual Comm. Lost | Injuries/Death | Camera Failure | Monitoring | Stop Motion |
| | Speech Comm. Lost | Injuries/Death | Microphones Failure | | Stop Motion |
| | Haptic Comm. Lost | | Inertial Sensor Failure | | Stop Motion |

High-level failures include starting from the localization to system communication losses, and their potential effects lead to sensor errors. Furthermore, the recommended actions manage the navigation system to prevent unrelated actions by stopping, slowing down, or triggering the emergency system. Low-level failures are described in Table A3.

**Table A3.** The low-level failure mode and effects analysis.

| Process Step | Potential Failure Mode | Potential Failure Effects | Potential Causes | Current Controls | Recomm. Action |
|---|---|---|---|---|---|
| | | | | **Low-Level Failures** | |
| Monitoring | Camera Failure | Safety System Error | Power Issue | Electric Design | H.I. + Shutdown |
| | | | Unplugged Cable | Reinforce connections | Stop Motion + H.I. |
| | | | Software Crash | Signal Monitoring | Restart Firmware |
| | | Visual Comm. Lost | Software Crash | None | Wait |
| | LiDAR Failure | Safety System Error | Power Issue | Electric Design | H.I. + Shutdown |
| | | | Unplugged Cable | Reinforce Connection | Stop Motion + H.I. |
| | | | Software Crash | Signal Monitoring | Restart Firmware |
| | GPS Failure | Unintended Excursion | Software Crash | Monitoring | Restart Firmware |
| | | | Signal Precision | None | Wait |
| | | Localization Imprecision | Satellites Comm. | Kalman Filter | Wait |
| | Odometry Failure | Localization Imprecision | Encoders Malfunction | Maintenance | Maintenance |
| | | Speeding | GPS Failures | Param. Optimization | Slow Down |
| | | Unintended Excursion | Software Crash | Monitoring | Restart Firmware |
| | | Aborted Execution | Localization Imprecision | Velocity Controller | H.I. |
| | WiFi Failure | App Comm. Lost | Low Reception | None | Wait |
| | | | Unplugged Cable | Reinforce connections | Stop Motion + H.I. |
| | | | Software Crash | Signal Monitoring | Restart Firmware |
| | Bluetooth Failure | App Comm. Lost | Low Reception | None | Wait |
| | | | Unplugged Cable | Reinforce connections | Stop Motion + H.I. |
| | | | Software Crash | Signal Monitoring | Restart Firmware |
| | Inertial Sensor Failure | Haptic Comm. Lost | External Noise | Kalman Filter | Wait |
| | | | Unplugged Cable | Reinforce connections | Stop Motion + H.I. |
| | | | Software Crash | Signal Monitoring | Restart Firmware |
| | Microphone Failure | Speech Comm. Lost | External Noise | None | Wait |
| | | | Unplugged Cable | Reinforce connections | Stop Motion + H.I. |
| | | | Software Crash | Signal Monitoring | Restart Firmware |

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
