# Peer review of "Towards Safe Robotic Agricultural Applications: Safe Navigation System Design for a Robotic Grass-Mowing Application through the Risk Management Method"

_robotics, doi:10.3390/robotics12030063_

Round 1
Reviewer 1 Report
Chapter 2 uses a lot of acronyms which are used in several places in the text without explaining them. All acronyms should be explained in parentheses when first used for better understanding.
All author drawn figures should use a consistent graphical style (e.g. compare figs. 2 and 6).
Figure 4. appears to be vertically compressed. Please redraw or correct.
Figure 8. has illegible text in sub-figures a, b and d - possible overlapping of images.
Chapter 9.1. Future work is quite short. It should be expanded.
You have a deep understanding of the research objective. It could be a good idea to make the article more understandable to readers who are not so familiar with the subject matter.
Author Response
Thanks a lot for your time and effort reviewing this paper.

Reviewer 2 Report
This paper describes initial steps for modeling and integrating a safety scheme based on FMEA for a grass-mowing application.
This topic is interesting and important, and the paper is well written, but some improvements must be made before publishing.
1. In the Related Work section, the research gap must be clarified. Authors must clearly explain what lacks in the literature, and afterwards how their proposed method answers the gap.
2. In section 5.3, the first sentence (lines 275-276) must include reference.
3. It is not clear how the human intervention is implemented for the safe action selection scheme. Where should be the human? How should he/she intervene? and more importantly, human reaction times and limitations such as fatigue must be considered.
4. Paragraph 7 should be renamed from "Experiments" to " Simulations", since no real experiments were conducted.
5. In figure 8, add explanations regarding what are the 'x' and the trajectories.
Author Response
Thanks a lot for your comments and suggestions. They definitely helped to improve the quality of the paper. I think most of the suggestions are attended in this set of modifications.

Reviewer 3 Report
Comment 1. The abbreviation that appears in the article should write its full name clearly when it first appears.
Comment 2. The article should add a literature review or additional references.
Comment 3. There is overlap in the contents of Table 4.
Comment 4. Lines 350-351: Is it reasonable to respawn a person after being collided with their speed and motion changing direction?
Comment 5. The parameters in Table 6 should be given corresponding explanations.
Comment 6. Line 372: The execution time does not match the data given in Table 6.
Comment 7. Please check for typographical and grammatical errors.
Comment 8. The Manuscript is too long and the charts and tables are irregular.
Comment 9. The Figure 9 don not belong to Conclusions part.
Author Response
Thanks a lot for your time and effort in reviewing the paper. I appreciate all the comments, trying to address as much as possible all the nice comments. Most of the suggestions have been fixed. Thanks again :)

Round 2
Reviewer 2 Report
Since authors have not provided a proper response file rather than a revised manuscript with the highlighted changes, it is very difficult to identify which comments were addressed and how. I think my comments 1-3 have not been addressed and therefore the paper requires a further revision.
1. In the Related Work section, the research gap must be clarified. Authors must clearly explain what lacks in the literature, and afterwards how their proposed method answers the gap.
2. In section 5.3, the first sentence (lines 275-276) must include reference.
3. It is not clear how the human intervention is implemented for the safe action selection scheme. Where should be the human? How should he/she intervene? and more importantly, human reaction times and limitations such as fatigue must be considered.
Author Response
My apologies if my last comments were short or inaccurate.
Research Gap:
It is true that this might look like any other paper that talks about obstacle detection and navigation systems, however, in agricultural applications there are not as many autonomous solutions fully released yet (outdoors, open-field, row-following, avoidance) even less the ones that take safety as a main problem to solve.
Also, there are navigation stacks (move_base, move_base_flex, nav2, etc) and perception solutions (human/obstacle detection, and tracking), but it is not so common to evaluate both solutions in a specific task (in this case grass cutting and harvesting).
In this paper, we wanted to show all the assumptions and considerations we set for an autonomous solution for agriculture to make it as safe as possible. We decide to use a Risk Management approach.
At the top, in the context of human-robot interaction so we decided to try the hardest scenario we could think about. As collisions are a main feature to evaluate the simulations provide a good start for this.
We consider that the research gap was described in the introduction, but I have added additional definition in the related work section
Human Intervention
As the core of our work is to make the robot as safe as possible, the Human Intervention is considered a part of our architecture that acts in cases where a robot state does not make sense (i.e. when a person changed from safe to lethal zone in just one frame). Instead of guessing or relying on the future states, the system asks for approval for resuming its tasks, in particular in cases where the system loses a person, for example when a person falls.
This intervention (aka approval) can be sent as a button signal, a web application, an app, or any other human-robot interface, which is one next work, btw.
Fatigue and reaction times are just relevant for emergency-stop sequence. Other subsystems are highly automized. However, we set the simulated human with different speeds so the fatigue could be analyzed in this way.
But I agree, human skills are important to be analyzed. It is a good idea to set each simulated person with a personality with: fatigue, robot reliability (how close he/she feels comfortable around the robot), and even, age, height, gender, physiological trait, etc. There are many factors to be considered, including Human Behaviors which is a very interesting and dense topic.
However, our work should not be considered as final results as mentioned in the conclusions. Safety must be improved further before having a final product in the field.
NOTE:
The paper writing was reviewed entirely. I commented on the paragraphs that have been reviewed.
Reviewer 3 Report
Thank you.
Author Response
Beforehand, I apologize for the inaccurate reply to the first comments you took the time to make.
Along the document, I have marked the paragraphs which have been modified to improve the story-telling and description of the work, in an attempt to provide a richer description of the approach.
What I realize now, and what you have told me before, is that I must provide a further description of the approach. I took more time to be more descriptive in several parts of the paper. Thanks again for all.
Round 3
Reviewer 2 Report
Authors have answered my comments and I therefore I recommend accepting the paper. I strongly advise authors to incorporate their answers of their private response to me into the manuscript, as it will increase the quality of the paper significantly. If these explanations regarding the research gap and human intervention will be added to the text, they will convince the audience of the Journal as they have convinced me. Good luck.